# Assessment of Condylar Changes in Patients with Degenerative Joint Disease of the TMJ After Stabilizing Splint Therapy: A Retrospective CBCT Study

**DOI:** 10.3390/diagnostics14202331

**Published:** 2024-10-19

**Authors:** Sara Steinbaum, Anabel Kelso, Nawal Firas Dairi, Normand S. Boucher, Wenjing Yu

**Affiliations:** 1Department of Orthodontics, School of Dental Medicine, University of Pennsylvania, Philadelphia, PA 19104, USA; 2School of Dental Medicine, University of Pennsylvania, Philadelphia, PA 19104, USA; 3Private Practice, 333 W Lancaster Ave, Wayne, PA 19087, USA

**Keywords:** temporomandibular joint, cone beam computed tomography, stabilizing splint therapy, condylar remodeling

## Abstract

Background/Objectives: Degenerative joint disease (DJD) of the TMJ can impact patients’ quality of life and complicate orthodontic treatment. Stabilizing splints are a common conservative treatment in managing TMDs symptoms, although their long-term effects on condylar morphology are poorly studied. This study aimed to assess the impact of stabilizing splints on condyle morphology using cone-beam computed tomography (CBCT) in patients with various stages of DJD. Forty-two condyles with pre- (T1) and post- (T2) splint therapy scans were analyzed. Methods: CBCT scans were sectioned into sagittal and coronal slices for condyle classification and measurement. T1 and T2 CBCTs were superimposed before linear and area measurements at different poles. Results: Our results indicate that condyles in the normal group remain unchanged after splint therapy. The majority of subjects in the degenerative groups remained in the same classification group: six out of fourteen degenerative-active patients became degenerative-repair, while three out of twenty-two degenerative-repair patients progressed to degenerative-active. There is no significant remodeling of condylar width pre- and post-splint therapy. On average, there is more bone deposition than reduction in condylar height after splint therapy, although individual variation exists. Conclusions: Stabilizing splints offer a low-risk intervention for managing DJD and may contribute to favorable adaptive changes in the condyles.

## 1. Introduction

Temporomandibular disorders (TMDs) encompass a variety of musculoskeletal and neuromuscular conditions involving the masticatory muscles, the temporomandibular joint (TMJ), and associated structures, and they are a significant source of chronic orofacial pain of non-dental origin [1]. According to the most recent literature, the incidence of TMDs in the global population is 34%, and the prevalence largely depends on geographical location [2]. Disorders of the TMJ can lead to considerable discomfort and dysfunction, impacting patients’ quality of life and complicating orthodontic treatment. Orthodontic treatment involves moving teeth, which can increase stress on the TMJ, potentially worsening existing joint disease [3,4]. A stable TMJ provides a foundation for effective and safe orthodontic treatment, minimizing the risk of aggravating the condition and ensuring the longevity of treatment results [5,6,7].

Degenerative joint disease (DJD) of the TMJ is prevalent and is characterized by bony changes such as sclerosis, flattening, and erosions of the condyle [8,9,10]. The etiology of degenerative changes in the TMJ includes decreased adaptive capacity of articulating structures and excessive or sustained physical stress [11]. Non-invasive treatments for TMDs, such as stabilizing splints (SS), are a standard treatment modality aimed at protecting the joint from overloading, reducing muscle hyperactivity, and mitigating articular strain from bruxism [12,13]. Occlusal splints offer a conservative treatment option to stabilize the TMJ before initiating orthodontic procedures [14]. Studies have demonstrated the efficacy of splint therapy in reducing pain and improving function in patients with TMDs, making it an essential first step for some orthodontic patients [13,15,16]. However, the literature lacks critical evaluations of the long-term effects of splint therapy on the disease process and morphology of the condyle. The present study aimed to assess the impact of stabilizing splint therapy on the morphology of the condylar head using cone-beam computed tomography (CBCT) in patients with various stages of degenerative joint disease. We hypothesize that (1) The defined state of the pathological process of condyle will be improved after SS therapy, indicative of a favorable adaptive process, and (2) the horizontal, vertical, and area measurements of condyle will increase in post-stabilizing splint CBCTs. The null hypothesis is that SS therapy does not change the bony classification of the condyle’s horizontal, vertical, or area measurements at each condylar pole based on pre- and post-treatment CBCTs.

## 2. Materials and Methods

### 2.1. Study Design

This retrospective study was approved by the Institutional Review Board (IRB) at the University of Pennsylvania (Protocol #853858). Forty-two condyles from twenty-one adult patients with significant TMD symptoms diagnosed with degenerative joint disease (DJD) of the TMJ were included. From May 2013 to May 2022, full-volume CBCT scans using an i-CATTM machine (KaVo Imaging, Hatfield, PA, USA) were obtained in the centric relation (CR) position with the patient’s mouth closed in the natural head position as part of routine orthodontic records. These CBCT images were taken at 120 kVp and 5 mA with a volume size of 16 × 13 cm^2^, exposure time of 3.7 s, and a voxel size of 0.3 mm [17]. Patients adhered to a regimen of full-time stabilizing splint wear (22–24 h per day), including daytime and nighttime appliances (Figure 1). The evening and nighttime appliance consisted of an occlusal platform covering the anterior and posterior teeth, fabricated with hard acrylic and adjusted to have an optimal distribution of posterior centric stops in the CR and shallow anterior guidance. It was worn for an average of 14 to 16 h a day. The daytime appliance consisted of occlusal platforms covering all posterior teeth to provide posterior centric stops but no anterior guidance. It was worn from before breakfast until after dinner, averaging 8 to 10 h daily. On average, patients had routine follow-up visits every four weeks throughout treatment to examine and adjust the occlusal appliances as needed. Patient compliance was reported and monitored during their visits. All patients also followed through with physical therapy; they were referred to a physical therapist with expertise in head and neck disorders contributing to TMD symptoms. Therapy varied depending on individual needs. Exercises were provided to address forward head posture, rounded shoulders, and tongue position at rest and during swallowing. Additional therapy focused on muscle relaxation and increasing range of motion was provided when needed. Stabilizing splints were removed when the patient reported significant improvement in TMD symptoms. Non-compliant patients and patients reporting a history of fracture, severe trauma, orthognathic surgery, or tumor were excluded from this study. Subjects meeting the criteria were included and organized by individual condyles. In total, 42 condyles with two sets of desired CBCT scans were included in this study. Ten subjects were excluded due to poor CBCT scan quality and/or incomplete records. By analyzing 42 condyles before (T1) and after (T2) splint treatment, we sought to provide insights into the effects of splint therapy before orthodontic treatment in DJD patients (Appendix A).

### 2.2. CBCT Measurement Protocol

All CBCT scans were de-identified, then oriented and analyzed using Dolphin Imaging 3D software (version 11.9, Dolphin Imaging & Management Solutions, Chatsworth, CA, USA). Each scan was oriented in three planes of space for frontal and lateral views. In the frontal view, the head was positioned with the orbital floor parallel to the ground. Frontal orientation was verified, and scans were laterally oriented to the Frankfort horizontal plane. After the initial orientation, the axial image displaying the greatest mediolateral dimension of the condyle was chosen. Reference lines were drawn to connect the medial and lateral poles of the condyle. Each joint was then sectioned into nine sagittal (each 2.0 mm thick) and coronal (each 1.0 mm thick) slices [17] (Figure 2A–E). The coronal and sagittal images of each condyle were randomized and blinded, and a senior orthodontist classified each condyle pre- and post-splint therapy as “normal condyle”, “degenerative-active”, or “degenerative-repair” using the guide from Phi et al. [18]. Briefly, the normal condyle has a normal size and shape. The osseous components are smooth, rounded, and without evidence of subchondral defects. The degenerative-active condyle demonstrates reduced size from the superior surface, loss of cortex along the anterosuperior surface with exposure of cancellous bone, and cavitation-type defects in the superior surface (erosion). The degenerative-repair condyle is small, and the condylar surface shows signs of flattening and consolidation of the erosive surface. The superior surface of the condyle is flat and corticated, or it is corticated and smooth and has an eburnated appearance. The opposing articular surfaces may form a congruent articulation.

After each T1 and T2 CBCT was initially oriented and slices were obtained for condyle classification, scans were prepared for 3D superimposition. Each T1 and T2 scan used four reference points (bilateral frontozygomatic suture and mental foramen) for superimposition (Figure 3A), and then they were initially superimposed on the cranial base (Figure 3B). Then, each CBCT was reoriented in a new configuration to best assess the condylar head in a true sagittal and coronal view. Laterally, scans were oriented horizontally to the widest anterior-posterior aspect of the condyle and vertically from Condylion to Gonion (Figure 3C). Then, from an inferior view, the condyles were oriented horizontally at the widest mediolateral condyle width (Figure 3D). The newly oriented T1 and T2 CBCTs were then superimposed on the mandible to assess changes in the condylar head before and after splint therapy. The internal symphysis and inferior alveolar nerve canal (IANC) were landmarks used to verify accurate mandibular superimposition (Figure 3E,F). Once superimposed, the medial, middle, and lateral poles were defined.

In the axial view, the widest mediolateral position of the T1 scan was selected, and the corresponding coronal line was set at the widest mediolateral position. The width of the condyle was calculated. The midway point was defined as the middle pole, and then the lateral and medial poles were defined as 4 mm from the middle pole. In the axial view, the line demarcating the specific pole remained constant as the sagittal and coronal views were analyzed (Figure 4A,B). In each sagittal view for each pole, a horizontal linear distance and an area measurement were obtained in both T1 and T2. In the coronal view for T1 and T2, vertical linear distances for the three poles and one coronal area were obtained for each condyle scan (Figure 4C).

Both T1 and T2 scans were re-oriented, and new superimpositions were obtained when analyzing the contralateral condyle. All measurements were taken without knowing the status of the scans. Horizontal, vertical, and area measurements were compared for each condyle at each time point. Randomized and blinded condyle classifications were compared at each time point. Both quantitative and qualitative assessments of condylar changes were analyzed for T1 and T2 at each medial, middle, and lateral pole.

### 2.3. Statistics

Statistical analysis was performed using SAS (version 9.4; SAS Institute Inc., Cary, NC, USA). Descriptive statistics were computed to determine the mean and standard deviations of the linear and area measurements at each medial, lateral, and middle pole. *t*-tests were calculated to compare the changes in linear measurements and area between T1 and T2 condyles. A *p*-value < 0.05 was considered statistically significant. The intraclass correlation coefficient (ICC) was calculated to assess both inter- and intra-reliability measurements between examiners and for each examiner at a two-week time point. The inter-reliability results were 0.976 for linear measurements and 0.997 for area measurements. The intra-reliability results ranged between 0.969–0.998 among the three examiners. All ICC results are indicative of excellent reliability [19]. A sample of 42 subjects achieved a desired power of 0.716. Power and effect size analyses were performed using G*Power (Version 3.1.9.6) (RID:SCR_013726) [20].

## 3. Results

### 3.1. Joint Diagnosis

A total of 42 condyles were analyzed and measured (Table 1). The pre-treatment T1 condyle classification revealed that four (9.5%) were “normal”, fourteen (33.3%) were “degenerative-active”, and twenty-four (57.2%) were “degenerative-repair” (Figure 5A). The post-treatment T2 condyle diagnosis revealed that four (9.5%) were “normal”, eleven (26.2%) were “degenerative-active”, and twenty-seven (65.3%) were “degenerative-repair” (Figure 5B). The four “normal” T1 condyles remained “normal” after splint therapy. Out of the fourteen “degenerative-active” at T1, eight condyles remained “degenerative-active”, and six condyles transitioned to “degenerative-repair”. Twenty-one out of the twenty-four T1 “degenerative-repair” condyles remained as “degenerative-repair”; however, three condyles transitioned from “degenerative-repair” to “degenerative-active” (Figure 5C).

### 3.2. Linear Changes

#### 3.2.1. Horizontal

From the sagittal view, a horizontal measurement was obtained at each medial, middle, and lateral pole before and after splint therapy. The average horizontal distance at the medial pole was 6.35 ± 1.53 mm at T1 and 6.28 ± 1.61 mm at T2. For the middle pole, the average horizontal distance was 6.74 ± 1.47 mm at T1 and 6.76 ± 1.59 mm at T2. For the lateral pole, the average horizontal distance was 6.71 ± 1.25 mm at T1 and 6.68 ± 1.29 mm at T2 (Table 2 and Figure 6A). To compare the dynamic change for each condyle before and after splint therapy, the difference between T1 and T2 was calculated. The average horizontal change was −0.07 ± 0.50 mm, 0.02 ± 0.47 mm, and −0.03 ± 0.70 mm at the medial, middle, and lateral poles, respectively (Table 3). Overall, there was no significant change in the horizontal dimension, and the majority of the subjects stayed stable without significant remodeling at all three poles (Figure 6B). We further divided each condyle at each pole based on the condylar classification at T1 into “normal (N)”, “degenerative-active (A)”, and “degenerative-repair (R)” groups. There was a slight increase in the control group at all three poles, no change in the degenerative-repair group, and a slight reduction in the degenerative-active group at the medial pole (Figure 7A–C). When we analyzed the data using the mean ± 1 standard deviation as our cut-off threshold to see how many subjects underwent bone deposition/resorption or stayed relatively stable during splint therapy, we found that the majority of the studied subjects were relatively stable, and in the degenerative-repair group, there was more resorption occurring in the medial and lateral poles compared to the middle pole. In the degenerative-active group, four condyles had bone deposition after splint therapy in the middle pole, and 3 condyles underwent bone resorption in the horizontal dimension at each pole. In the normal group, three out of four condyles stayed relatively stable, and one condyle had bone deposition at the medial and lateral poles while all four condyles remained stable at the middle pole (Table 4).

#### 3.2.2. Vertical

From the coronal view, a vertical measurement was obtained at each pole. The average vertical distance at the medial pole was 7.13 ± 1.55 mm at T1 and 7.37 ± 1.58 mm at T2. For the middle pole, the average vertical distance was 7.48 ± 1.51 mm at T1 and 7.67 ± 1.48 mm at T2. For the lateral pole, the average horizontal distance was 6.63 ± 1.63 mm at T1 and 6.73 ± 1.73 mm at T2 (Table 2 and Figure 6C). There was a significant difference (*p* = 0.033, dz = 0.153) between T1 and T2 at the medial pole for the vertical change, with an average difference of 0.24 ± 0.70 mm. The average vertical change at the middle and lateral poles was 0.19 ± 0.63 mm and 0.10 ± 0.80 mm, respectively, with a bone deposition tendency (Table 3 and Figure 6D). Overall, there was more change in the vertical dimension compared to the horizontal dimension, indicating that the condyles have more remodeling activity in the vertical dimension during splint therapy. After breaking down the data into subgroups, only the degenerative–active group exhibited an overall bone resorption tendency with negative changes between T1 and T2 (Figure 7D–F). The headcount number also showed that more condyles underwent bone deposition than resorption during splint therapy in all groups at each pole. In the degenerative-repair group, more than two-thirds of condyles remained stable in the vertical dimension at each pole; six out of twenty-four condyles at the medial and middle poles and five at the lateral pole had bone deposition, while two condyles at the medial pole and one at the middle and lateral poles had bone resorption. The results of the degenerative-active group were similar in the medial and middle poles, but three condyles underwent bone deposition and resorption separately. In the normal group, no condyle had bone resorption, and all four remained stable at the lateral pole, while one at the medial pole and two at the middle pole sxhibited significant bone deposition after splint therapy (Table 4).

### 3.3. Area Changes

#### 3.3.1. Sagittal Area

The average sagittal area at the medial pole was 45.54 ± 14.95 mm^2^ at T1 and 47.28 ± 15.30 mm^2^ at T2. At the middle pole, the sagittal area was 47.28 ± 15.36 mm^2^ at T1 and 50.04 ± 15.96 mm^2^ at T2, with a statistically significant difference (*p* = 0.009, dz = 0.143). At the lateral pole, it was 40.19 ± 12.19 mm^2^ at T1 and 41.90 ± 13.24 mm^2^ at T2 (Table 2 and Figure 6E). The most change between T1 and T2 occurred in the area measurements, with the change at the middle pole offering the most statistical significance, and the changes at medial and lateral poles were almost statistically significant and worth mentioning (*p* = 0.058 and 0.051, dz = 0.116 and 0.134). The average sagittal area change between T1 and T2 was 1.74 ± 5.76 mm^2^, 2.24 ± 5.33 mm^2^, and 1.71 ± 5.52 mm^2^ at the medial, middle, and lateral poles, respectively (Table 3 and Figure 6G). If we further break the data down into subgroups, we can see that the degenerative-repair group and normal group both had positive changes at all three poles. In contrast, the degenerative-active group was close to the neutral line (Figure 7G–I). The headcount number also showed a similar trend when compared to the vertical linear changes. In the degenerative-repair group, more than two-thirds of condyles remained stable in the sagittal area at each pole; five out of twenty-four condyles at the medial and lateral poles and six at the middle pole had increased area, while two condyles at the middle and lateral poles, and one at the medial pole had decreased area. The results of the degenerative-active group showed three condyles that underwent bone deposition and resorption separately. In the normal group, no condyle had bone resorption, and all four remained stable at the lateral pole. In contrast, one at the medial pole and two at the middle pole had significant bone deposition after splint therapy (Table 4).

#### 3.3.2. Coronal Area

One coronal area measurement was obtained for each condyle at T1 and T2, encompassing all the poles. The average coronal area was 119.2 ± 33.01 mm^2^ at T1 and 122.9 ± 32.68 mm^2^ at T2 (Table 2 and Figure 6F). The difference in coronal area between T1 and T2 was almost statistically significant (*p* = 0.051, dz = 0.113), with an average increase of 3.72 ± 11.98 mm^2^ (Table 3 and Figure 6G). After breaking down the data into subgroups, we could observe that all three groups had positive changes after splint therapy, ordered as degenerative-active, degenerative-repair, and normal from the least to the most (Figure 7J). The headcount number showed a similar pattern among the three groups. More than 65% of the studied condyles remained stable in the coronal area measurement, while five out of twenty-four condyles in the degenerative-repair group, three out of fourteen in the degenerative-active group, and one out of four in the normal group had increased coronal area. There were two subjects in each of the degenerative-repair and degenerative-active groups who had a coronal area reduction. In contrast, condyles in the control group were relatively stable, with no significant reduction (Table 4).

## 4. Discussion

Our results revealed a distribution of condylar classifications that showed some shifts post-treatment. Specifically, the proportion of normal condyles remained unchanged, and there was no significant bone reduction in the normal condyle subgroup at all dimensions, including linear and area measurements, indicating that stabilizing splint treatment may be a conservative option that has little effect on stable condylar morphology. The result is supported by a systematic review that assessed the efficacy of various splint therapies, including stabilizing splints. They found that while these therapies are beneficial in managing TMDs symptoms, they do not significantly alter the function or structure of a healthy TMJ [16]. Conversely, among the condyles diagnosed as degenerative-active at T1, 57% remained in the same category, while 43% transitioned to a degenerative-repair state. This shift suggests that stabilizing splints may offer a therapeutic effect in promoting the repair process in some actively degenerating condyles.

Similarly, a previous study also concluded that patients who underwent splint therapy had less bone destruction compared to a control who did not undergo splint therapy [14]. For condyles initially categorized as degenerative-repair, the majority (88%) remained in this category, with a small percentage (12%) regressing to the degenerative-active state. This finding indicates that while splint therapy may stabilize the condition of condyles undergoing repair, there is a small subset that may experience a relapse or progression of degeneration.

The analysis of dimensional changes at each condylar pole provided further insight into the morphological adaptations occurring during the treatment period. The medial pole showed a significant vertical change, with an average increase of 0.24 mm. This average vertical increase suggests that the splint treatment may facilitate vertical adaptation or stabilization in the medial aspect of the condyle. The sagittal area changes, particularly in the middle pole (average change of 2.24 mm^2^), highlight a significant remodeling potential in this region. With stabilizing splints, the stress stimuli are changed in the TMJ area, which may impact the osteo-immune microenvironment and lead to the remodeling of the condylar head more towards bone deposition rather than degeneration. Although the horizontal changes at all poles were minimal, the vertical linear and sagittal area changes may indicate a more complex three-dimensional remodeling process influenced by splint therapy. The SS therapy period varies from 12–24 months since the splint therapy was concluded when patients reported significant improvement in TMD symptoms, and this wide therapy range might also introduce variability in the results. Overall, the relatively minimal changes in all dimensions align with previous studies, reinforcing the potential use of stabilizing splints as a conservative, low-risk treatment option for patients with degenerative joint disease.

According to Ok et al., when examining condylar remodeling after splint therapy, the greatest proportion of bone formation post-splint occurred in the anterior segments, with the anterior medial locations exhibiting the highest proportion at 67%. This corresponds with our findings showing that the vertical dimension at the medial pole had the most statistically significant difference post-treatment [14]. Furthermore, it is reported that the predominant areas for bone formation after stabilizing splint therapy in patients with osteoarthritis were postero-medial, lateral, medial, and posterior-superior [6]. In the literature, there is one study on condylar mediolateral width and condylar height changes after splint therapy, but it did not find any statistically significant differences for these parameters [21].

The main limitation of this study was that the results were not compared to a non-treatment control. Unfortunately, a control group with a similar diagnosis placed under observation with no treatment is not ethically justified, so we do not have a control population. Without a control group, it is difficult to attribute these changes solely to the treatment, since the passage of time can also result in a favorable adaptive response. Nevertheless, anecdotally, this patient population reported a significant reduction in their TMD symptoms and improved quality of life. Additionally, this study was retrospective in nature. Due to the relatively small sample size, the power analyses were not desirable if we broke down our subjects into potential confounding factors, such as age, sex, and initial disease severity. Future better-designed prospective studies with controlled age, sex, initial DJD severity, and duration of SS therapy are needed to develop a more comprehensive understanding of the changes SS therapy may have on condylar morphology and DJD progression.

## 5. Conclusions

The findings from this study suggest that stabilizing splint therapy may contribute to favorable adaptive changes in the condyles of patients with degenerative joint disease of the TMJ, and the small dimensional changes have resulted in significant reductions in TMD symptoms and improved quality of life. Specifically, full-time splint therapy with supportive physical therapy may promote a shift from active degeneration to a repair phase and facilitate favorable morphological changes, particularly in the medial and middle poles. However, the observed variability in condylar response highlights the need for personalized treatment approaches and further research to understand the long-term effects and mechanisms underlying splint-induced adaptations. Ultimately, stabilizing splint therapy offers a low-risk intervention for managing DJD in the TMJ, contributing to improved patient outcomes.

## Figures and Tables

**Figure 1 diagnostics-14-02331-f001:**
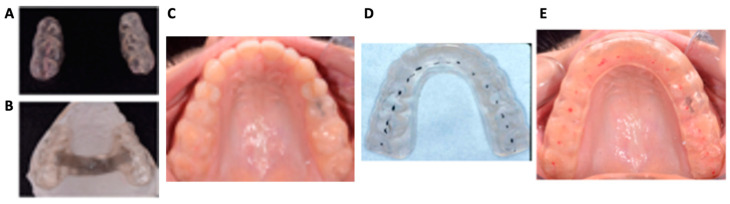
Stabilizing splints used in this study. (**A**–**C**) Examples of daytime appliances: A daytime appliance designed with separate posterior splints (**A**) and a thin palatal connector (**B**). Occlusal view of separate posterior splints fitted in the patient’s maxillary dentition (**C**). (**D**,**E**) Examples of nighttime appliances fitted in the patient’s maxillary arch showing bilateral balanced occlusion.

**Figure 2 diagnostics-14-02331-f002:**
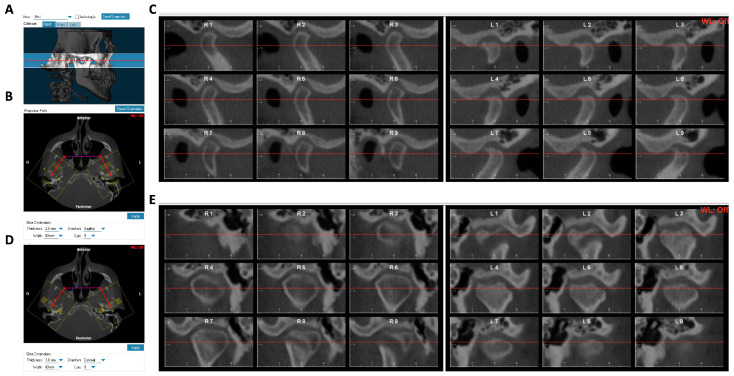
Generation of sagittal and coronal condyle cuts from CBCT for pre- and post-splint condyle classification. (**A**) X-ray built from TMJ area. (**B**,**C**) The screenshots from Dolphin show how sagittal cuts were obtained from CBCT scans. (**D**,**E**) The screenshots from Dolphin show how coronal cuts were obtained from CBCT scans.

**Figure 3 diagnostics-14-02331-f003:**
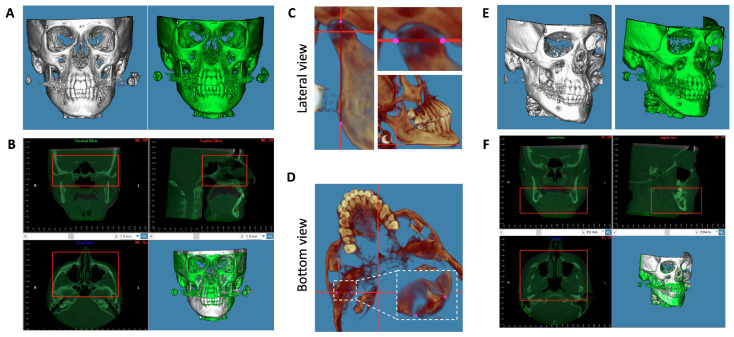
T1 and T2 mandibular superimposition and orientation procedure. (**A**,**B**) The initial cranial base orientation position and superimposition. Four initial reference points (bilateral frontozygomatic suture and mental foramen) were used for T1 and T2 superimpositions (**A**). The result of the initial cranial base superimposition for orienting T1 and T2 using structures within the red boxes (**B**). (**C**,**D**) The reorientation process to best assess the condylar head. (**E**,**F**) The final mandibular orientation position and superimposition using structures within the red boxes, where measurements were then obtained.

**Figure 4 diagnostics-14-02331-f004:**
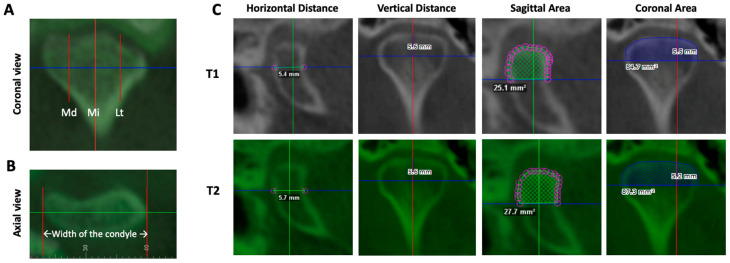
Examples of how horizontal, vertical, and area measurements were obtained. (**A**,**B**) Coronal and axial views of the condyle. To define the lateral, middle, and medial poles of the condyle, set the axial line (blue) at the widest aspect of the coronal condyle (**A**) and then set the coronal line (green) at the widest aspect of the axial condyle (**B**). The width of the condyle was measured from the axial view, and the midway point was defined as the middle (Mi) pole, and then the lateral (Lt) and medial (Md) poles were defined as 4 mm from the middle pole, as shown in (**A**). (**C**) Examples of the measurements of horizontal distance, vertical distance, sagittal area, and coronal area from the superimposed T1 and T2 middle pole.

**Figure 5 diagnostics-14-02331-f005:**
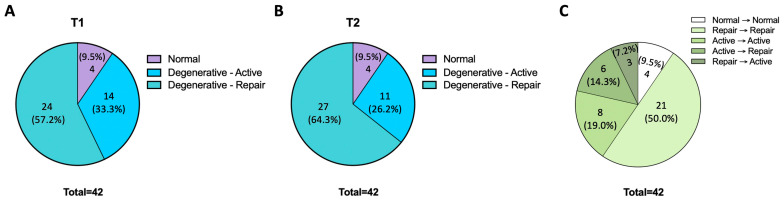
Distribution of T1 and T2 condyle classifications. (**A**,**B**) The pie chart shows the number and percentage for each condyle classification at T1 (**A**) and T2 (**B**). (**C**) The pie chart shows the number and percentage for each T1 to T2 transition subset.

**Figure 6 diagnostics-14-02331-f006:**
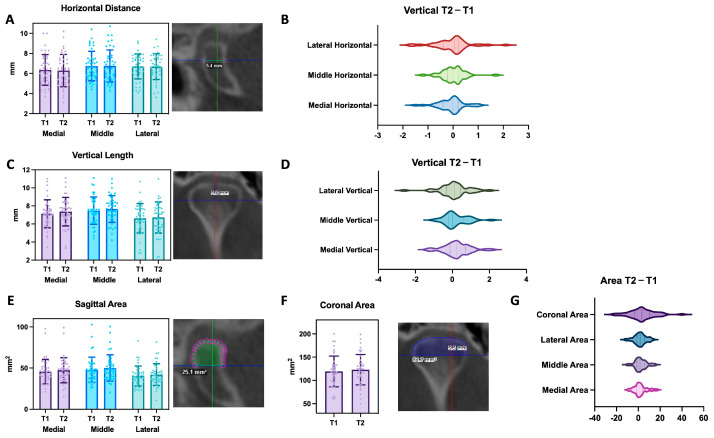
The T1 and T2 comparison between measurements of horizontal and vertical distances and sagittal and coronal areas at each medial, middle, and lateral pole. (**A**) The horizontal linear measurement comparison between T1 and T2 at each pole. Each dot represents one subject. The bar represents the mean, with lines representing the SD. (**B**) The violin plot shows the distribution of the dynamic change of each pole from T1 to T2. Positive numbers represent increased horizontal linear distance at T2 when compared to T1. (**C**) The vertical linear measurement comparison between T1 and T2 at each pole. Each dot represents one subject. The bar represents the mean, with the lines representing the SD. (**D**) The violin plot shows the distribution of the dynamic change of each pole from T1 to T2. Positive numbers represent increased vertical linear distance at T2 when compared to T1. (**E**) The sagittal area measurement comparison between T1 and T2 at each pole. Each dot represents one subject. The bar represents the mean, with the lines representing the SD. (**F**) The coronal area measurement comparison between T1 and T2 at each pole. Each dot represents one subject. The bar is at mean with lines for SD. (**G**) The violin plot shows the distribution of the dynamic change of each pole from T1 to T2. Positive numbers represent increased area measurements at T2 when compared to T1.

**Figure 7 diagnostics-14-02331-f007:**
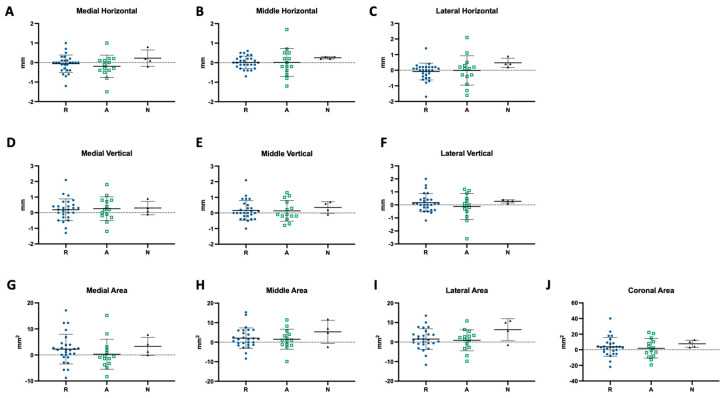
The changes between T1 and T2 measurements of horizontal and vertical distances and sagittal and coronal areas at each medial, middle, and lateral pole. (**A**–**J**) Scatter dot plots of each measurement. Lines represent the mean with SD. Each dot represents one subject.

**Table 1 diagnostics-14-02331-t001:** The distribution of condyle classifications.

	Normal	Degenerative		Total
		Active	Repair	
T1	4 (9.5%)	14 (33.3%)	24 (57.2%)	42 (100%)
T2	4 (9.5%)	11 (26.2%)	27 (64.3%)	42 (100%)
T2 Maintained	4 (9.5%)	8 (19%)	21 (50%)	33 (78.5%)
T2 Changed	0	3 (7.2%)	6 (14.3%)	9 (21.5%)

**Table 2 diagnostics-14-02331-t002:** The descriptive statistics of measurements of horizontal and vertical distances and sagittal and coronal areas at each medial, middle, and lateral pole at T1 and T2.

	Horizontal						Vertical						Area							
	Medial Pole (mm)		Middle Pole (mm)		Lateral Pole (mm)		Medial Pole (mm)		Middle Pole (mm)		Lateral Pole (mm)		Medial Sagittal (mm^2^)		Middle Sagittal (mm^2^)		Lateral Sagittal (mm^2^)		Coronal (mm^2^)	
	T1	T2	T1	T2	T1	T2	T1	T2	T1	T2	T1	T2	T1	T2	T1	T2	T1	T2	T1	T2
Minimum	3.70	3.10	4.40	4.30	4.10	3.60	3.40	3.30	4.40	4.20	2.80	2.30	20.90	15.20	25.90	20.20	22.20	10.50	60.20	50.40
Maximum	10.00	10.20	10.40	10.70	9.20	9.40	11.00	11.10	11.10	11.00	10.70	11.00	97.90	99.20	102.9	100.5	83.10	81.70	200.0	198.9
Range	6.30	7.10	6.00	6.40	5.10	5.80	7.60	7.80	6.70	6.80	7.90	8.70	77.00	84.00	77.00	80.30	60.90	71.20	139.8	148.5
Mean	6.35	6.28	6.74	6.76	6.71	6.68	7.13	7.37	7.48	7.67	6.63	6.73	45.54	47.28	47.80	50.04	40.19	41.90	119.2	122.9
SD	1.53	1.61	1.47	1.59	1.25	1.29	1.55	1.58	1.51	1.48	1.63	1.73	14.95	15.30	15.36	15.96	12.19	13.24	33.01	32.68
*p*-value	0.378		0.821		0.81		0.033*		0.061		0.444		0.058		0.009*		0.051		0.051	
Effect size dz	0.045		0.013		0.024		0.153		0.127		0.060		0.116		0.143		0.134		0.113	

**Table 3 diagnostics-14-02331-t003:** The descriptive statistics of the dynamic changes between the T1 and T2 of horizontal and vertical distances and sagittal and coronal areas at each medial, middle, and lateral pole.

	Horizontal			Vertical			Area			
	Medial Pole (mm)	Middle Pole (mm)	Lateral Pole (mm)	Medial Pole (mm)	Middle Pole (mm)	Lateral Pole (mm)	Medial Sagittal (mm^2^)	Middle Sagittal (mm^2^)	Lateral Sagittal (mm^2^)	Coronal (mm^2^)
Minimum	−1.50	−1.20	−1.70	−1.30	−1.00	−2.60	−8.80	−9.90	−11.70	−22.00
Maximum	1.00	1.70	2.10	2.10	2.10	2.00	17.10	15.30	13.50	40.00
Range	2.50	2.90	3.80	3.40	3.10	4.60	25.90	25.20	25.20	62.00
Mean	−0.07	0.017	−0.03	0.24	0.19	0.10	1.74	2.24	1.71	3.72
SD	0.50	0.47	0.70	0.70	0.63	0.80	5.76	5.33	5.52	11.98

**Table 4 diagnostics-14-02331-t004:** The number of subject condyles that underwent bone deposit/resorption or stayed stable during splint therapy in different subgroups.

		Medial Pole			Middle Pole			Lateral Pole			Coronal		
		Repair	Active	Norm	Repair	Active	Norm	Repair	Active	Norm	Repair	Active	Norm
Horizontal	Deposition	3	0	1	2	4	0	1	1	1			
	Maintain	17	11	3	21	7	4	20	10	3			
	Resorption	4	3	0	1	3	0	3	3	0			
Vertical	Deposition	6	5	1	6	4	2	5	3	0			
	Maintain	16	8	3	17	8	2	18	8	4			
	Resorption	2	1	0	1	2	0	1	3	0			
Area	Deposition	5	2	1	6	3	2	5	2	3	5	3	1
	Maintain	18	11	3	16	10	2	17	10	1	17	9	3
	Resorption	1	1	0	2	1	0	2	2	0	2	2	0

## Data Availability

The data presented in this study are contained within this article and Appendix A.

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
