# Peer review of "Assessment of Condylar Changes in Patients with Degenerative Joint Disease of the TMJ After Stabilizing Splint Therapy: A Retrospective CBCT Study"

_diagnostics, 2024, doi:10.3390/diagnostics14202331_

Round 1

Reviewer 1 Report

Comments and Suggestions for Authors

Thank you for the opportunity to review. My comments are listed below:

  1. I don’t understand why the document submitted for review still contains comments like 'Commented [SS1]: Split sentence in two.'
  2. Keywords: - Abbreviations like 'TMJ' and 'CBCT' should not be included in the keywords.
  3. 'Stabilizing' – Please standardize the capitalization, either use all capital letters or all lowercase letters.
  4. Line 30 – 'Temporomandibular disorder (TMD)' – The correct abbreviation is TMDs, according to the new guidelines.
  5. Line 30-35 – These are not up-to-date data, the authors refer to studies from 2021 and 2008. In scientific papers, references should be made to recent literature. The latest meta-analysis indicates that the incidence of TMDs in the global population is 34%, and the prevalence largely depends on geographical location. Please revise this section and refer to the study 10.3390/jcm13051365.
  6. Line 40 – '[5-8]' – A maximum of 3 citations is sufficient to support a statement – this comment applies to the entire paper.
  7. Line 54 – Add a hypothesis and an anti-hypothesis.
  8. L55-56 – This fits better in the methods section, not here.
  9. L62-79 – Is this the authors' original methodology? If not, please provide citations.
  10. All the graphics in the text are of very poor quality. Please improve them.
  11. Statistical analysis – Please add sample size calculations - 10.1002/jgf2.600.
  12. Statistical analysis – In the context of statistical analysis, please include effect size calculations for each p-value result - 10.1001/jamaoto.2023.0159.
  13. In the results, please add a 0 before the decimal point so that it's '0.997' instead of '.997'. This applies to the entire text.
  14. I am unable to evaluate the results and conclusions of the paper without the sample size and effect size analyses. After adding these, I will provide further analysis.
  15. L388 – Add page numbers.
Comments on the Quality of English Language

I am not a language specialist, but this paper is difficult to read. I suggest a language review.

Reviewer 2 Report

Comments and Suggestions for Authors

This paper analyzes the effects of stabilizing splints on condyle morphology in patients with temporomandibular joint degenerative joint disease.

There are some observations to be:

Materials and Methods:

There's no discussion of informed consent from patients. This should be addressed.

The limited sample may not be representative of the broader population of patients with temporomandibular joint degenerative joint disease, potentially affecting the external validity of the findings.

There's no mention of how compliance was monitored or ensured.

The post-treatment time point (T2) varies from 12-24 months. Could this wide range introduce variability in the results?

The patients underwent physical therapy, but no details are provided.

It's unclear if the researchers performing the measurements were also blinded to the pre/post-treatment status of the scans.

There's no mention of how image quality was assessed or if any scans were excluded due to poor quality.

The guide for classifying condyles as "normal," "degenerative-active," or "degenerative-repair" is mentioned but not described.

Consider discussing and controlling for potential confounding factors such as age, sex, severity of initial DJD, and duration of symptoms.

The images are of extremely poor quality.

Results:

Again, all figures are of extremely poor quality.

Discussion:

The discussion implies causal relationships between splint therapy and observed changes. However, without a control group, it's difficult to attribute these changes solely to the treatment. This limitation should be more explicitly addressed.

The discussion doesn't mention the small sample size, which is a significant limitation.

More emphasis on the clinical implications of the observed changes must be addressed. For instance, how do the small dimensional changes translate to patient outcomes or quality of life? Potential confounding factors such as subjects age, sex, or severity of initial condition, which could influence treatment outcomes should be discussed.

Round 2

Reviewer 1 Report

Comments and Suggestions for Authors

Thank you for resubmitting the paper for review.

  1. Most of the comments have been addressed. However, the manuscript still requires some minor corrections. Firstly, citations should be in the format ‘[1]’. This needs to be corrected throughout the text.
  2. Line 82, 87, 355, 374, 386 – correct the abbreviation ‘TMD’ to ‘TMDs’.
  3. Remove the unnecessary bold formatting from ‘Table 4’ and ‘Table 2 and Figure 6E’. This applies to the entire text.
  4. Lines 265, 288, 292, 311 – add the effect size calculations.

Best regards.

Author Response

We appreciate the reviewer's diligent evaluation of this manuscript, and your insightful comments have significantly contributed to the enhancement of the quality of this work. Please find our responses as below:

Comment 1: Most of the comments have been addressed. However, the manuscript still requires some minor corrections. Firstly, citations should be in the format ‘[1]’. This needs to be corrected throughout the text.

Response 1: Thank you for your comment. We’ve changed the citation format followed by the MDPI Reference List and Citations Style Guide.

Comment 2: Line 82, 87, 355, 374, 386 – correct the abbreviation ‘TMD’ to ‘TMDs’.

Response 2: Thank you for your comment. We’ve corrected the abbreviation to TMDs accordingly in Line 82, 87, 355, 374, and 384, we don’t find TMD abbreviation in Line 386.

Comment 3: Remove the unnecessary bold formatting from ‘Table 4’ and ‘Table 2 and Figure 6E’. This applies to the entire text.

Response 3: Thank you for your advice. We removed all the bold formatting for all the tables  and figures in the text.

Comment 4: Lines 265, 288, 292, 311 – add the effect size calculations.

Response 4: Thank you for your comment. We have added the effect size calculations into the text accordingly.

Thank you once again for your attention to our research.

Reviewer 2 Report

Comments and Suggestions for Authors

No more questions.

Author Response

We appreciate the reviewer's diligent evaluation of this manuscript, and your insightful comments have significantly contributed to the enhancement of the quality of this work.

Thank you once again for your attention to our research.